# Sex- and Co-Mutation-Dependent Prognosis in Patients with SMARCA4-Mutated Malignancies

**DOI:** 10.3390/cancers15102665

**Published:** 2023-05-09

**Authors:** Minggui Pan, Chen Jiang, Zheyang Zhang, Ninah Achacoso, Aleyda V. Solorzano-Pinto, Pam Tse, Elaine Chung, Jennifer Marie Suga, Sachdev Thomas, Laurel A. Habel

**Affiliations:** 1Department of Oncology and Hematology, Kaiser Permanente, Santa Clara, CA 94051, USA; 2Division of Research, Kaiser Permanente, Oakland, CA 94612, USA; chen.x.jiang@kp.org (C.J.); ninah.s.achacoso@kp.org (N.A.); aleyda.v.solorzano-pinto@kp.org (A.V.S.-P.); pamela.tse@kp.org (P.T.); elaine.chung@kp.org (E.C.); laurel.habel@kp.org (L.A.H.); 3Division of Oncology, Stanford University School of Medicine, Stanford, CA 94304, USA; 4State Key Laboratory of Cellular Stress Biology, School of Life Sciences, Faculty of Medicine and Life Sciences, Xiamen University, and National Institute for Data Science in Health and Medicine, Xiamen University, Xiamen 361102, China; zzhang@stu.xmu.edu.cn; 5Department of Oncology and Hematology, Kaiser Permanente, Vallejo, CA 94589, USA; jennifer.m.suga@kp.org (J.M.S.); sachdev.p.thomas@kp.org (S.T.)

**Keywords:** SMARCA4, NGS, TP53, KRAS, STK11, CDKN2A, Keap1, prognosis

## Abstract

**Simple Summary:**

SMARCA4-mutated tumors are associated with poor prognosis. However, it is not clear if male and female patients have the same or different prognosis and if additional common mutations within the tumor play a role in prognosis. We have found that male patients had substantially worse prognosis than female patients whose tumor carried a SMARCA4 mutation. In addition, different co-existing mutations including the mutation of *TP53*, *KRAS*, *CDKN2A*, STK11, and *Keap1* were associated with differential prognosis. Our study provides helpful insight for prognostic stratification in clinical practice and for understanding the pathogenic mechanism of this unique subtype of malignancy.

**Abstract:**

Background: Whether sex and co-mutations impact prognosis of patients with SMARCA4-mutated (mutSMARCA4) malignancies is not clear. Methods: This cohort included patients from Northern California Kaiser Permanente with next-generation sequencing (NGS) performed from August 2020 to October 2022. We used Cox regression modeling to examine the association between sex and overall survival (OS), adjusting for demographics, performance status, Charlson comorbidity index, receipt of treatment, tumor mutation burden (TMB), and *TP53*, *KRAS*, *CDKN2A*, *STK11*, and *Keap1* co-mutations. Results: Out of 9221 cases with NGS performed, 125 cases (1.4%) had a mutSMARCA4. The most common malignancies with a mutSMARCA4 were non-small cell lung cancer (NSCLC, 35.2%), esophageal and stomach adenocarcinoma (12.8%), and cancer of unknown primary (11.2%). The most common co-mutations were *p53* (mutp53, 59.2%), *KRAS* (mutKRAS, 28.8%), *CDKN2A* (mutCDKN2A, 31.2%), *STK11* (mutSTK11, 12.8%), and *Keap1* (mutKeap1, 8.8%) mutations. Male patients had substantially worse OS than female patients both among the entire mutSMARCA4 cohort (HR = 1.71, [95% CI 0.92–3.18]) with a median OS of 3.0 versus 43.3 months (*p* < 0.001), and among the NSCLC subgroup (HR = 14.2, [95% CI 2.76–73.4]) with a median OS of 2.75 months versus un-estimable (*p* = 0.02). Among all patients with mutSMARCA4, mutp53 versus wtp53 (HR = 2.12, [95% CI 1.04–4.29]) and mutSTK11 versus wtSTK11 (HR = 2.59, [95% CI 0.87–7.73]) were associated with worse OS. Among the NSCLC subgroup, mutp53 versus wtp53 (HR = 0.35, [0.06–1.97]) and mutKRAS versus wtKRAS (HR = 0.04, [0.003-.45]) were associated with better OS, while mutCDKN2A versus wtCDKN2A (HR = 5.04, [1.12–22.32]), mutSTK11 versus wtSTK11 (HR = 13.10, [95% CI 1.16–148.26]), and mutKeap1 versus wtKeap1 (HR = 5.06, [95% CI 0.89–26.61}) were associated with worse OS. Conclusion: In our cohort of patients with mutSMARCA4, males had substantially worse prognosis than females, while mutTP53, mutKRAS, mutCDKN2A, mutSTK11 and mutKeap1were differentially associated with prognosis among all patients and among the NSCLC subgroup. Our results, if confirmed, could suggest potentially unidentified mechanisms that underly this sex and co-mutation-dependent prognostic disparity among patients whose tumor bears a mutSMARCA4.

## 1. Introduction

*SMARCA4* (*Brg1*) is a core component of SWI/SNF (the yeast switch in mating type [SWI]/sucrose nonfermentation [SNF] complex) chromatin remodeling complex that plays critical roles in the development of many tissues and organs by regulating chromatin accessibility and gene expression, DNA repair, cell cycle control and others [1,2,3]. SMARCA4 protein contains ATPase activity and is a catalytic subunit of SWI/SNF complex. Homozygous deletion of *SMARCA4* causes embryonic lethality at peri-implantation stage while its heterozygous deletion causes increased risk of neoplasia [4]. The extremely rare germline mutations of *SMARCA4* cause small cell carcinoma of ovary with hypercalcemia type (SCCOHT) and uterine sarcoma [5,6,7]. Mutations of *SMARCA4*, though relatively uncommon, appear to be present across a large spectrum of malignancies [7,8]. Analysis of genomic studies have found approximately 20% of malignancies harbor an alteration within the genes of SWI/SNF complex [9,10]. Experimentally, inactivation of *SMARCA4* potentiates lung adenocarcinoma and its metastasis [11,12], while displacement of *SMARCB1* (BAF47), another key component of SWI/SNF complex, from SWI/SNF complex by SSX-SS18 fusion protein is associated with oncogenesis of synovial sarcoma [13]. Clinically, absence of BAF47 (INI-1) expression is characteristic of all epithelioid sarcomas [14]. 

The roles of SMARCA4 in human malignancies appear broad and complex. A landmark study by Fernando et al. using data from genomic profiling of more than 130,000 solid tumors identified more than 9000 patients whose cancer bore one or more mutSMARCA4 [15]. Several thousands of *SMARCA4* variants were identified. The most common malignancies that bore a mutSMARCA4 were non-small cell lung cancer (NSCLC) and cancer of unknown primary (CUP) in this study [15]. This and other studies have shown that patients whose tumor bore a mutSMARCA4 have poor prognosis [15,16,17]. 

In this study we performed survival analysis of a cohort of 125 patients with advanced malignancies that bore a mutSMARCA4. Patients were enrollees of Northern California Kaiser Permanente (KPNC) and identified by routine next-generation sequencing (NGS). We aimed to examine the association of sex and the most common co-mutations including *TP53* (mutTP53), *KRAS* (mutKRAS), *CDKN2A* (mutCDKN2A), *STK11* (mutSTK11) and *Keap1* (mutKeap1) mutations with OS.

## 2. Methods

### 2.1. Study Population

Our analytic dataset included KPNC patients with locally advanced or metastatic cancer with StrataNGS (Strata Oncology, Ann Arbor, Michigan) performed from August 2020 to October 2022 who were identified as having mutSMARCA4. We assembled patient data on demographics, Charlson comorbidity index (CCI), performance status (PS), anatomic primary, and receipt of chemotherapy from the electronic medical record (Epic) and cancer registry database. CCI was based on the 12 month-period prior to diagnosis of locally advanced or metastatic cancer. This study was approved by the KPNC institutional review board with waiver of consent.

### 2.2. StrataNGS

StrataNGS of advanced malignancies was initiated in November 2017 in KPNC and and is currently a 429-gene, pan-solid tumor, NGS assay for formalin-fixed, paraffin-embedded tumor tissue, performed on co-isolated DNA and RNA [18,19]. *SMARCA4 was* included into StrataNGS panel in August 2020. Among 125 patients with a SMARCA4 mutation, 67 patients (53.6%) had a missense *SMARCA4* mutation, 14 (11.2%) patients had a *SMARCA4* deep deletion, 6 (4.8%) patients had an in-frame deletion of 1–5 amino acids, 20 (16%) patients had a frameshift, 17 patients (13.6%) had a *SMARCA4* splice site mutation, and 1 patient (0.8%) had a duplication.

### 2.3. Treatment

Patients who received any single or combination of treatments including chemotherapy, targeted therapy, and immune checkpoint inhibitors were considered exposed to a treatment. Three patients with a mutSMARCA4 had *EGFR* exon 19 deletion (2 male and 1 female) and one male patient had a *ROS1* fusion gene. All four patients had a response to a tyrosine kinase inhibitor (Osimertinib for *EGFR* mutation and entrectinib for *ROS1* fusion gene). Four out of 49 patients who were treated with an immune checkpoint inhibitor (either pembrolizumab or nivolumab) had a partial response. One female patient who had Lynch syndrome developed a CUP (likely gynecologic origin) that bore a mutSMARCA4 and had a partial response to pembrolizumab. Another patient with Lynch syndrome and colon cancer that also bore a mutSMARCA4 responded to pembrolizumab as well.

### 2.4. Statistical Analysis

OS was measured from the date of diagnosis of locally advanced or metastatic cancer to the date of death or end of study follow-up (2 February 2023), whichever came first. We used Pearson’s chi squared test to assess differences in distributions of demographics and *TP53*, *KRAS*, *CDKN2A, STK11* and *Keap1* co*-*mutations and the one-way ANOVA test to assess differences in continuous variables. We performed unadjusted (univariate) OS analysis and estimated median OS using Kaplan-Meyer plot (log rank test). The number of patients at risk in the Kaplan-Meyer OS curves accounted for left truncation. Time since diagnosis of advanced cancer was the time scale used in the regression models, allowing for delayed entry into the cohort at the time of receipt of NGS results (i.e., left truncation, with study entry ranging from 0 to 7.0 years post-diagnosis, median 0.5 months.) [20]. We used Cox proportional hazards regression models for estimating hazard ratios (HRs) and 95% confidence intervals (CI) for the association between mutation subsets and OS, adjusted for covariates. Covariates included in our main regression models (and unless otherwise stated) were age (continuous), sex (male, female), ethnicity (Non-Hispanic White, Black, Asian, Hispanic, other/unknown), PS (0 to 1, 2 to 4), CCI (continuous), treatment received (yes, no), smoking status (yes, no), *TP53* [yes, no], *KRAS* [yes, no], *CDKN2A* [yes, no], *STK11* [yes, no] mutation, *Keap1* [yes, no]. The statistical analysis was performed using SAS software version 9.4, R (R Core Team, 2020).

## 3. Results

### 3.1. Demographics

Out of 9221 patients (among them 1989 patients had the diagnosis of lung cancer) with NGS results, 125 cases (1.4%) had a mutSMARCA4. All 125 patients were included into this study. The most common cancer types among patients with a mutSMARCA4 included NSCLC (35.2%), esophageal/gastric adenocarcinoma (12.8%), CUP (11.2%) and colon cancer (9.6%) (Table 1). There were 60 female (48.0%) and 65 male patients (52.0%) (Table 2). Compared to female patients, male patients had a higher median CCI (1 vs. 0), a higher percent had PS of 2–4 (52.0% vs. 31.6) or were current smokers (24.0% vs. 15.8%), a lower percent received anti-cancer treatment (61.5% vs. 76.7%), and a higher percent carried mutp53 (70.8% vs. 46.7%). Because of small numbers, many differences were not statistically significant. Among the subgroup of patients with NSCLC (Table 3), there were 19 (43.2%) female patients and 25 (56.8%) male patients. In this subgroup, a higher percent of males than females had PS 2–4 or had a p53 mutation, although numbers were again small. Males and females appeared to be fairly similar with respect to smoking history and the distribution of other co-mutations.

### 3.2. Male Patients Had Substantially Worse OS Compared to Female Patients

Using Cox regression modeling adjusting for demographics, PS, CCI, treatment received and co-mutations, male patients had worse OS than female patients (HR = 1.71, [95% CI 0.92–3.18]) among all patients with mutSMARCA4 (Figure 1A). The median OS for male patients was 3.0 months while 43.8 months for female patients (*p* < 0.001) (Figure 1B). Among the NSCLC subgroup, male patients also had substantially worse OS than female patients (HR = 14.2, [95% CI 2.76–73.40]) (Figure 1A), with median OS of 2.75 months for male patients while un-estimable for female patients (*p* = 0.02) (Figure 1C). We downloaded and performed analysis of all available TCGA cases (total 10,182 cases). There were 448 (4.5%) cases who had SMARCA4 mutation/variant. Among these 448 cases, 34 had advanced stage, 13 were female patients and 21 were male patients and there was no significant OS difference between male and female patients. However, the median OS for both male and female patients was more than 30 months (Appendix A), which is very different than our cohort and the largest cohort that was previously reported [15].

### 3.3. Impact of Co-Mutations on Prognosis

Using Cox regression modeling and adjusting for demographics, PS, CCI, receipt of treatment and co-mutations, among all patients with mutSMARCA4, mutp53 versus wtp53 was associated with worse OS (HR = 2.11, [95% CI 1.04–4.29]) (Figure 2A), with median OS of 4.2 versus 12.9 months (*p* = 0.03, Figure 2B). MutKRAS versus wtKRAS and mutCDKN2A versus wtCDKN2A were not associated with substantial OS difference (Figure 2A). MutSTK11 versus wtSTK11 was associated with substantially worse OS (HR = 2.59, [95% CI 0.87–7.73]). Surprisingly, among the NSCLC subgroup, mutp53 versus wtp53 (HR = 0.35, [95% CI 0.06–1.97]) and mutKRAS versus wtKRAS (HR = 0.04, [95% CI 0.003-.45]) were associated with better OS (Figure 2A), while mutCDKN2A versus wtCDKN2A (HR = 5.04, [95% CI 1.12–22.32]), mutSTK11 versus wtSTK11 (HR = 13.10, [95% CI 1.12–148.26]) and mutKeap1 versus wtKeap1 (HR = 5.06, [95% CI 0.89–26.61]) were associated with worse OS (Figure 2A). The HR for mutTP53 versus wtTP53 with the NSCLC subgroup had a wide confidence limit. The sample size of each subgroup in the NSCLC cohort was relatively small especially with mutSTK11 (n = 9) and mutKeap1 (n = 8). We found that patients whose tumor had both mutSMARCA4 and mutKRAS and an additional mutTP53 had worse OS compared to those patients whose tumor retained wtTP53 (HR = 16.53, [95% CI 1.86–147.11]). Interestingly, among 16 patients with mutSTK11 in the entire cohort, only 2 patients had mutTP53, and among 9 patients with mutSTK11 in the NSCLC cohort, only 1 patient had mutTP53, suggesting that mutTP53 and mutSTK11 could be mutually exclusive in patients with a mutSMARCA4 (Table 4).

We examined the impact of co-mutations within the subgroup of patients who received immune checkpoint inhibitors. Within this group of 49 patients, mutTP53 versus wtTP53 was associated with worse prognosis (HR = 3.98, [95% CI 1.14–13.91), and mutCDKN2A versus wtCDKN2A was also associated with worse prognosis in a lesser degree with wide confidence intervals (HR = 2.09, [95% CI 0.57–7.61]). MutKRAS was not associated with substantial difference in OS, while mutSTK11 was surprisingly associated with better OS (HR = 0.55, [95% CI 0.08–3.55] (Figure 2C) with wide confidence limits too.

## 4. Discussion

Our study with 125 patients with a mutSMARCA4 has revealed several interesting findings: First, male patients had substantially worse OS than female patients among the entire cohort and among the subgroup with NSCLC. Second, mutp53 versus wtp53 was associated with worse OS among the entire cohort, but better OS among the subgroup with NSCLC. Third, mutKRAS versus wtKRAS was not associated with substantially different OS among the entire cohort but was associated with better OS among the subgroup with NSCLC. Fourth, mutCDKN2A versus wtCDKN2A was not associated with substantially different OS among the entire cohort but was associated with substantially worse OS among patients with NSCLC. Finally, mutSTK11 versus wtSTK11 and mutKeap1 versus wtKeap1 were associated with worse OS among the entire cohort and the subgroup with NSCLC.

It is not surprising that male patients had worse OS than female patients both among all patients with mutSMARCA4 and among the NSCLC subgroup. However, the magnitude of OS difference is surprising among both groups. The median OS of approximately 3 months for male patients among all mutSMARCA4 patients and among the NCSLC subgroup, compared to the median OS of female patients of approximately 43 months among all mutSMARCA4 patients and un-estimable among the NSCLC subgroup, suggests that the poor prognosis associated with *SMARCA4*-mutated malignancies was primarily driven by the poor prognosis of male patients. In the study by Schoenfeld et al. male patients had modestly worse OS compared to female patients among a cohort of lung cancer patients with and without *SMARCA4* mutation [21], which is different than our cohort that included only *SMARCA4*-mutated patients. Our sample size with NSCLC subgroup is also relatively small which could potentially exaggerate the difference. Data extracted from the Surveillance, Epidemiology and End Results Database for 36 cancers by sex and age for the period 1977 to 2006 showed higher mortality with males than females in multiple cancers including cancer of lips, hypopharynx, larynx, esophagus, etc. [22,23]. In certain malignancies such as bladder, colon and breast cancer, male patients were found to have better OS than female patients in some studies [24,25,26]. Mechanistically, why male patients could have such a dramatically worse OS than female patients remains not clear and may warrant further investigations. *SMARCA4* has been found in most studies to function as a tumor suppressor gene, but in some studies, it appears to play oncogenic roles on a context-dependent manner [27,28,29,30,31,32,33]. For example, it was shown that *Brg1* (*SMARCA4*) inhibits dedifferentiation of pancreatic acinar cells prior to their neoplastic transformation while promotes proliferation of pancreatic ductal adenocarcinoma cells by maintaining mesenchymal-like gene expression [27]. In another study, expression of SMARCA4 protein appears required for Wnt-associated tumorigenesis of murine small intestine [30]. The spectrum of *SMARCA4* mutations in human cancers is large and different mutations appear to lose its ATPase catalytic activities or possess different residual activities [15]. Some *SMARCA4* mutations appear to possess dominant-negative properties [34]. In our entire cohort of patients, male patients appear to have worse PS, higher CCI and higher percent with a history of smoking. Smoking, and possibly other unmeasured lifestyle factors, might have led male patients to have worse PS and CCI at the time of diagnosis of the *SMARCA4*-mutated malignancy and could be partially responsible for their worse OS by accelerating cancer progression even before their diagnosis. Though this correlation was not as evident in the patients with NSCLC only. Other unidentified mechanisms likely exist. The small TCGA cohort of patients with SMARCA4 mutation/variant likely does not represent the typical patients with SMARCA4-mutated malignancies. It is possible that some of the SMARCA4 variants that were detected were not pathogenic. It is also possible different SMARCA4 mutation/variant and different histology confer different prognosis as reported by Fernando et al. [15].

More surprisingly, mutp53 was associated with worse OS among all patients in the entire cohort, but not among patients with NSCLC only. Mutp53 was previously shown to be associated with worse prognosis in patients with NSCLC [35,36,37]. Our study suggests that mutp53 may be associated with a different mechanism in influencing prognosis among patients with a mutSMARCA4. Such a mechanism may operate differently depending on different histologic type. It may cause accelerated progression of most malignancies but not NSCLC. Previous studies showed that SMARCA4 knockdown caused replicative stress and decreased p21 expression by reducing the binding of p53 to its promoter, suggesting that functional loss of both SMARCA4 and p53 may increase tumor vulnerability to therapy [38,39]. Similarly surprising was the results with mutKRAS being associated with substantially better OS among the NSCLC subgroup in our study. MutKRAS is an established negative factor for the prognosis of NSCLC based on a meta-analysis and individual studies [40,41]. However, the natural history of patients with these co-mutations may be changing due to the advances with immune checkpoint inhibitors (ICIs) [42,43,44,45]. Some studies showed that concurrent KRAS and TP53 mutations were associated with high level of PD-L1 and better response to (ICIs) in patients with lung adenocarcinoma [43,45]. Our results showing that mutCDKN2A was associated with worse OS among patients with NSCLC was not surprising, as a previous study with early-stage NSCLC showed that mutCDKN2A was associated with worse prognosis [46]. A separate study showed that mutCDKN2A was associated with resistance to immune checkpoint inhibitors in patients with NSCLC [47]. Palbociclib, a CDK4/6 inhibitor, was found to have only modest activity in NSCLC patients with a *CDKN2A* mutation [48]. STK11 is a serine and threonine kinase and a tumor suppressor gene, its germline inactivating mutation is associated with Peutz-Jeghers syndrome [49]. Somatic mutations of *STK11* are most frequently identified in NSCLC. There is evidence mutSTK11 could be associated with resistance to immune checkpoint inhibitors [50]. A retrospective study on 60 patients with a *STK11* mutation showed mutSTK11 was associated with worse progression-free survival (PFS) and OS [51]. Our data surprisingly showed that within the subgroup of patients who received immune checkpoint inhibitor therapy, mutSTK11 was associated with better OS, though among the entire cohort and NSCLC only, mutSTK11 was associated with substantially worse OS. Another retrospective study showed that patients with metastatic cancer bearing both mutTP53 and mutSTK11 had a 21% 15-month PFS compared to 0% of patients whose cancer retained wtTP53 but bore mutSTK11 [52], suggesting that mutTP53 versus wtTP53 was associated with better PFS among patients with mutSTK11, consistent with our finding that mutTP53 was not associated with worse OS among NSCLC patients. In a study with 77 patients with NSCLC bearing a *STK11* mutation, it was found that *KRAS* mutation exacerbated OS while *TP53* mutation conferred better OS, though the sample size with different co-mutations in this study was small [44]. The mechanisms of how these common co-mutations interact with *SMARCA4* mutation in different cellular contexts to control cell proliferation and differentiation remain unclear. It is possible that the biological actions of mutTP53 and mutKRAS led to altered tumor microenvironment that is more sensitive to treatment in tumors that carried a SMARCA4 mutation. The study by Xue et al. showed that expression of cyclin D1 (CCND1) was reduced in patients with SMARCA4-mutated lung cancer which led to synthetic lethality with CDK4/6 inhibition [53]. Despite overall inhibition of CDK4/6 has not been shown to be effective in lung cancer, it is possible the subset of patients with a SMARCA4 mutation may be more sensitive to CDK4/6 inhibitors [48]. The prevalence of *STK11* mutation in patients with a mutSMARCA4 appears similar to that of all NSCLC patients, but *TP53* co-mutation appears to be mutually exclusive. This is not surprising as *SMARCA4* and *STK11* are both tumor suppressor genes. *Keap1* mutation was associated with worse prognosis in both the entire cohort and the NSCLC subgroup, consistent with the study by Schoenfeld [21].

This study has several strengths. Our dataset with the entire cohort is relatively large for patients with a mutSMARCA4 which is relatively uncommon. Second, all patients received comprehensive primary and specialty services from a large integrated healthcare system that consists of 21 medical centers. Our study included five co-mutations in multivariate OS analyses in addition to adjusting for other appropriate factors. Our study also has limitations. First, it is a retrospective study and some patients did not have StrataNGS performed at the time of diagnosis of advanced disease until months later. Nonetheless, we used appropriate statistical methods to address this issue [20]. Second, the sample size of the NSCLC subgroup was small, leading to wide confidence limits for some estimates.

In conclusion, our study indicated that male patients had substantially worse prognosis than female patients among all patients with mutSMARCA4 and among the NSCLC subgroup, suggesting that poor prognosis of *SMARCA4*-mutated malignancies observed in other studies may be primarily driven by male patients. In addition, different co-mutations appear to be associated with different prognosis among the entire cohort and among the NSCLC subgroup. Our results, if confirmed, could suggest potentially unidentified mechanisms that underly the sex disparity and the interactions of the common genomic alterations with *SMARCA4* mutation and could be helpful for prognostic stratification in clinical practice.

## Figures and Tables

**Figure 1 cancers-15-02665-f001:**
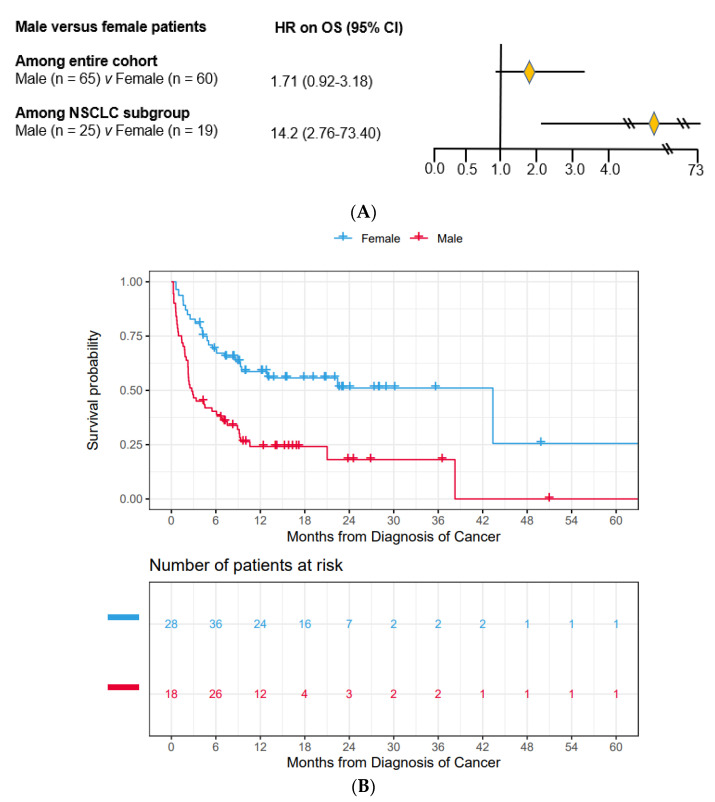
(**A**). Hazard ratios (HR) of OS for male versus female patients among the entire cohort and among patients with NSCLC only by Cox regression modeling. CI, confidence interval. (**B**). Kaplan-Meyer curves of OS for male versus female patients among the entire cohort. The median OS was 3.0 months for male patients and 43.8 months for female patients (*p* < 0.001). The number of patients at risk accounted for left-truncation. (**C**). Kaplan-Meyer curves of OS for male versus female patients among the NSCLC subgroup. The number of patients at risk accounted for left-truncation. The median OS was 2.75 months for male patients and un-estimable for female patients (*p* = 0.02).

**Figure 2 cancers-15-02665-f002:**
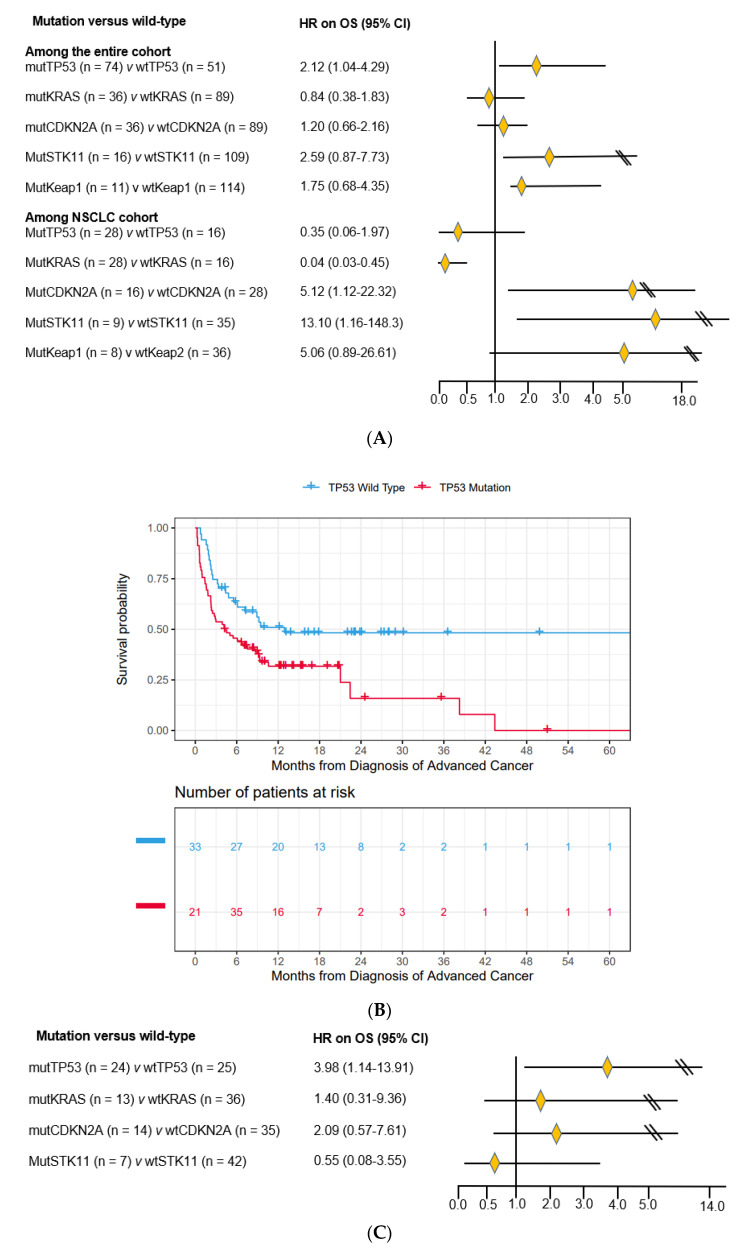
(**A**). Hazard ratio of OS for patients with mutation (mut) versus wild-type (wt) of the most common genomic alterations among the entire cohort and among patients with NSCLC only. Mutp53, *p53* mutation; mutKRAS, *KRAS* mutation; mutCDKN2A, *CDKN2A* mutation; mutSTK11, *STK11* mutation; mutKeap1, *Keap1* mutation. (**B**). Kaplan-Meier OS curves for patients with mutTP53 versus wtTP53 among the entire cohort. The number of patients at risk accounted for left-truncation. Median OS was 4.2 versus 12.9 months, *p* = 0.03. (**C**). Hazard ratio of OS for patients with mutation (mut) versus wild-type (wt) among the subgroup of patients (n = 49) who received immune checkpoint inhibitor treatment. Mutp53, *p53* mutation; mutKRAS, *KRAS* mutation; mutCDKN2A, *CDKN2A* mutation; mutSTK11, *STK11* mutation.

**Table 1 cancers-15-02665-t001:** Histology distribution of patients with a SMARCA4 mutation.

	SMARCA4 (n = 125)
Ampullary cancer (%)	2 (1.6)
Bladder cancer (%)	3 (2.4)
Breast cancer (%)	5 (4.0)
Cancer of unknown primary (%)	14 (11.2)
Colon cancer (%)	12 (9.6)
Endometrial cancer (%)	6 (4.8)
Esophageal/stomach cancer (%)	16 (12.8)
Gallbladder/cholangiocarcinoma (%)	3 (2.4)
Head and neck cancer (%)	1 (0.8)
Kidney cancer (%)	1 (0.8)
Non-small cell lung cancer (%)	44 (35.2)
Melanoma (%)	3 (2.4)
Ovarian cancer (%)	3 (2.4)
Pancreatic cancer (%)	5 (4.0)
Prostate cancer (%)	2 (1.6)
Skin squamous cell carcinoma (%)	2 (1.6)
Sarcoma (%)	2 (1.6)
Small cell lung cancer (%)	1 (0.8)

Notes: all 5 cases with breast cancer were negative for BRCA1/2 mutation. All 4 cases with BRCA1 mutation had a SMARCA4 mutation, and 2 of them had stomach cancer, 1 had colon cancer and 1 had skin SCC. Of 4 cases with BRCA2 mutation, 2 had SMARCA4 mutation (unknown primary and lung adeno with exon 19 deletion) and 2 had SMARCB1 mutation (1 colon cancer and 1 bladder cancer).

**Table 2 cancers-15-02665-t002:** Demographics of female and male patients with a *SMARCA4* mutation.

		Females (n = 60)	Males (n = 65)	*p*-Value
Age		68 (62–92)	67 (65–75)	0.56
Race	White	34 (58.6)	44 (68.8)	0.60
	Black	6 (10.3)	4 (6.2)	
	Hispanic	4 (6.9)	5 (7.8)	
	Asian	14 (24.1)	11 (17.2)	
	Unknown	2 (3.3)	1 (1.5)	
PS	0–1	30 (50.0)	26 (40.0)	0.12
	2–4	20 (33.3)	33 (50.8)	
	Unknown	10 (16.7)	6 (9.2)	
CCI		0 (0–10)	1 (0–9)	0.004
Smoking status	Current smoker	4 (6.7)	8 (12.3)	0.07
	Past smoker	29 (48.3)	40 (61.5)	
	Never smoker	27 (45.0)	17 (26.2)	
TMB > 10		23 (38.3)	25 (38.5)	0.98
Treatment	Yes	46 (76.7)	40 (61.5)	0.07
	No	14 (23.3)	25 (38.5)	
MutTP53		28 (46.7)	46 (70.8)	0.006
MutKRAS		17 (28.3)	19 (29.2)	0.91
MutCDKN2A		16 (26.7)	20 (30.8)	0.61
MutSTK11		9 (15.0)	7 (10.8)	0.48
Keap1		6 (10)	5 (7.7)	0.65

The numbers inside the parenthesis represent percentage except for age and CCI. TMB, tumor mutation burden. MutTP53, TP53 mutation; MutKRAS, KRAS mutation; MutCDKN2A, CDKN2A mutation; MutSTK11, STK11 mutation; MutKeap1, Keap1 mutation. PS, performance status; CCI, Charlson Comorbidity Index. TMB, tumor mutation burden. For the categorical variables, we used the Pearson’s Chi-squared test and for the continuous variable we used one-way ANOVA test.

**Table 3 cancers-15-02665-t003:** Demographics of male versus female patients with non-small cell lung cancer (NSCLC) with a *SMARCA4* mutation.

		Females (n = 19)	Males (n = 25)	*p*-Value
Age		65 (46–96)	70 (54–83)	0.93
Race	White	13 (68.4)	16 (64.0)	0.72
	Black	3 (15.8)	2 (8.0)	
	Hispanic	1 (5.3)	2 (8.0)	
	Asian	2 (10.5)	5 (20)	
PS	0–1	8 (42.1)	11 (44.0)	0.08
	2–4	6 (31.6)	13 (52.0)	
	Unknown	5 (26.3)	1 (4.0)	
CCI		0 (0–10)	0 (0–9)	0.32
Smoking status	Current smoker	3 (15.8)	6 (24.0)	0.54
	Past smoker	15 (78.9)	16 (64.0)	
	Never smoker	1 (5.3)	3 (12.0)	
TMB > 10		9 (47.4)	14 (56.0)	0.58
Treatment	Yes	13 (68.3)	15 (60.0)	0.56
	No	6 (31.6%)	10 (40.0)	
MutTP53		10 (52.6)	18 (72.0)	0.18
MutKRAS		11 (57.9)	17 (68.0)	0.49
MutCDKN2A		7 (36.8)	9 (36.0)	0.95
MutSTK11		5 (26.3)	4 (16.0)	0.40

The numbers inside the parenthesis represent percentage except for age and CCI. TMB, tumor mutation burden. MutTP53, TP53 mutation; MutKRAS, KRAS mutation; MutCDKN2A, CDKN2A mutation; MutSTK11, STK11 mutation. PS, performance status; CCI, Charlson Comorbidity Index. TMB, tumor mutation burden. For the categorical variables, we used the Pearson’s Chi-squared test and for the continuous variable we used one-way ANOVA test.

**Table 4 cancers-15-02665-t004:** Prevalence of *STK11* mutation (mutSTK11) among all patients and among NSCLC patients with a *SMARCA4* mutation.

	Among All Patients		Among NSCLC Only	
	mutTP53	wtTP53	mutTP53	wtTP53
mutSTK11 n = 16 (%)	2 (12.5)	14 (87.5)	1 (6.25)	15 (93.75)
*p* value	<0.001		<0.001	

## Data Availability

Kaiser Permanente Northern California (KPNC) Institutional Review Board has not provided approval for StrataNGS data on individual patients used in this study to be placed in a public access repository. However, researchers can request access to use this study data by contacting the DOR Data Sharing Workgroup at DOR-DataSharingWorkgroup@kp.org.

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
