# Peer review of "Sex- and Co-Mutation-Dependent Prognosis in Patients with SMARCA4-Mutated Malignancies"

_cancers, 2023, doi:10.3390/cancers15102665_

Round 1
Reviewer 1 Report
Comments for authors:
In this manuscript, the authors tried to study sex- and co-mutation-dependent prognosis in patients with SMARCA4-mutated malignancies. The authors show few interesting findings. However, the discussion needs to be modified to explain the interesting results clearly and convincingly.
1. How many SMARCA4 mutations were found in the cohort? Missense or nonsense mutations? Locations?
2. In the methods, the authors mentioned to isolate DNA and RNA. Did the authors use DNA or RNA as template for NGS test?
3. Results – Demographics, 125 cases/9251 patients = 0.013556 = 1.4%, not 1.3%.
4. Did authors subtype non-small cell lung cancer? Are they adenocarcinoma, squamous cell carcinoma, or other types of lung carcinoma?
5. Abbreviations in the tables and figures should be spelled out under tables or figures, for example PS, CCI, etc.
6. Statistical analysis methods should be mentioned under tables or figures.
7. What are the possible mechanisms that male patients had substantially worse OS than female patients?
8. Why was mutp53 associated with better OS among NSCLC? Generally, mutp53 is associated with worse OS in many malignancies.
9. Why was mutKRAS associated with different OS among different types of malignancies and better OS of NSCLC?
10. Did the authors compare patients with both mutp53 and mutSTK11, either mutp53 or mutSTK11, or without mutp53 and mutSTK11 for OS among NSCLC?
11. Did the authors compare patients with both mutp53 and mutKRAS, either mutp53 or mutKRAS, or without mutp53 and mutKRAS for OS among NSCLC?
12. In the discussion, the authors discussed the potential effects of immune checkpoint inhibitors on OS in patients with the mutations, however, the authors did not show data if there are effects of immune checkpoint inhibitors on OS of patients with individual mutation or concurrent multiple mutations. Can the authors obtain and analyze the data?
Author Response
Response to reviewer #1 comment
Pan et al.
Reviewer #1:
Comments for authors:
In this manuscript, the authors tried to study sex- and co-mutation-dependent prognosis in patients with SMARCA4-mutated malignancies. The authors show few interesting findings. However, the discussion needs to be modified to explain the interesting results clearly and convincingly.
Response: We thank the reviewer for the thorough review and constructive comment and suggestions.
- How many SMARCA4 mutations were found in the cohort? Missense or nonsense mutations? Locations?
Response: We have added information under StrataNGS in the Methods section. Among 125 patients with a SMARCA4 mutation, 67 patients (53.6%) had a missense SMARCA4 mutation, 14 (11.2%) patients had a SMARCA4 deep deletion, 6 (4.8%) patients had an in-frame deletion of 1-5 amino acids, 20 (16%) patients had a frameshift, 17 patients (13.6%) had a SMARCA4 splice site mutation, and 1 patient (0.8%) had a duplication.
- In the methods, the authors mentioned to isolate DNA and RNA. Did the authors use DNA or RNA as template for NGS test?
Response: Both isolated DNA and RNA were used for the NGS test (please see Method under StrataNGS section).
- Results – Demographics, 125 cases/9251 patients = 0.013556 = 1.4%, not 1.3%.
Response: We have made the change as suggested.
- Did authors subtype non-small cell lung cancer? Are they adenocarcinoma, squamous cell carcinoma, or other types of lung carcinoma?
Response: We have added this info now. Among 44 patients with NSCLC, 41 had adenocarcinoma, 2 had NSCLC without known subtype and 1 had sarcomatoid NSCLC (please see Demographics under Result section).
- Abbreviations in the tables and figures should be spelled out under tables or figures, for example PS, CCI, etc.
Response: We have added the abbreviations as suggested un the tables (2 and 3) and figures.
- Statistical analysis methods should be mentioned under tables or figures.
Response: We have added statistical method under the tables and figures.
- What are the possible mechanisms that male patients had substantially worse OS than female patients?
Response: The mechanisms are not clear but may be related in part to lifestyle like smoking. We have added this discussion in the second paragraph of the Discussion section: “In our entire cohort of patients, male patients appear to have worse PS and CCI and higher percent with history of smoking. The smoking lifestyle possibly caused male patients to have worse PS and CCI at the time of diagnosis of the SMARCA4-mutated malignancy and could be partially responsible for their worse OS by accelerating cancer progression even before their diagnosis. Though this correlation was not as evident in the patients with NSCLC only. Other unidentified mechanisms likely exist.”
- Why was mutp53 associated with better OS among NSCLC? Generally, mutp53 is associated with worse OS in many malignancies.
Response: The mechanism of this is not clear. We have added additional discussion for this (Previous studies showed that SMARCA4 knockdown caused replicative stress and decreased p21 expression by reducing the binding of p53 to its promoter, suggesting that functional loss of both SMARCA4 and p53 may increase tumor vulnerability to therapy). Please see 3rd paragraph under Discussion section.
- Why was mutKRAS associated with different OS among different types of malignancies and better OS of NSCLC?
Response: We are not clear about the mechanism of this observation but have discussed it in the Discussion section. Please see 3rd paragraph—(Some studies showed that concurrent KRAS and TP53 mutations were associated with high level of PD-L1 and better response to (ICIs) in patients with lung adenocarcinoma. We have added additional discussion: “The mechanisms of how these common co-mutations interact with SMARCA4 mutation in different cellular contexts to control cell proliferation and differentiation remain unclear. It is possible that the biological actions of mutTP53 and mutKRAS led to altered tumor microenvironment that is more sensitive to treatment in tumors that carried a SMARCA4 mutation”.
- Did the authors compare patients with both mutp53 and mutSTK11, either mutp53 or mutSTK11, or without mutp53 and mutSTK11 for OS among NSCLC?
Response: This is an excellent suggestion. However, among 16 patients with STK11 mutation, only 2 patients had TP53 mutation, therefore it is not feasible to perform a comparison analysis. (Please see Table 4).
- Did the authors compare patients with both mutp53 and mutKRAS, either mutp53 or mutKRAS, or without mutp53 and mutKRAS for OS among NSCLC?
Response: This is an excellent suggestion too. Among 36 patients with mutKRAS, 14 patients also had a TP53 mutation and 22 with wtTP53. MutTP53 was associated with worse OS in this subgroup of patients (please see revised Figure 2A in the Result section).
- In the discussion, the authors discussed the potential effects of immune checkpoint inhibitors on OS in patients with the mutations, however, the authors did not show data if there are effects of immune checkpoint inhibitors on OS of patients with individual mutation or concurrent multiple mutations. Can the authors obtain and analyze the data?
Response: There were 49 patients who received immune checkpoint inhibitor treatment in all patients. We have performed a Cox regression modeling analysis on this subgroup of patients and found mutTP53, mutCDKM2A were associated with worse OS, but mutSTK11 was associated with better OS, despite among the entire cohort and NSCLC subgroup, mutSTK11 was actually associated with worse OS. We have provided this piece of additional data in Figure 2C (new Figure) and added additional discussion (3rd paragraph in the Discussion section). Thanks for the suggestion.
Reviewer 2 Report
The Article addressed whether sex and co-mutations impact prognosis of patients with SMARCA4-11 mutated (mutSMARCA4) malignancies, a topic that was previously is not clear. This cohort included patients with next-generation sequencing (NGS) data available.
Male patients had substantially worse OS than female patients both among the entire mutSMARCA4 cohort (HR = 1.71, [95% CI .92-3.18]) with a median OS of 3.0 versus 43.3 months (P < .001), and among the NSCLC subgroup (HR = 14.2, [95% CI 2.76-73.4]) with a median OS of 2.75 months versus un-estimable (P = .02). The reported cohort of patients with mutSMARCA4, males had substantially worse prognosis than females, while mutTP53, mutKRAS, mutCDKN2A and mutSTK11 were differentially associated with prognosis among all patients and among the NSCLC subgroup. The results could suggest potentially unidentified mechanisms that underly this sex and co-mutation-dependent prognostic disparity among patients whose tumor bears a mutSMARCA4.
The study is well written and overall ready for publication.
Comments:
1) The sample sizes are small (especially tumor when separated by tumor type). Would it be possible to extrapolate from public data ( such as the TCGA) to further look at these co-alterations and sex disparities?
2) SMARCA4-mutated male patients had substantially dramatic worse prognosis than female patients among all patients with mutSMARCA4 and among the NSCLC subgroup. The mechanistic aspect of sex disparity should be further discussed, whether it may be lifestyle, hormonal or X-Chromosome linked.
3) Mutant p53 was higher among males ( again can this be seen in larger public data sets)
4) Manuscript “ SMARCA4 loss is synthetic lethal with CDK4/6 inhibition in non-small cell lung cancer” (PMID: 30718506) is relevant and should be discussed in discussion. This study suggests that SMARCA4 mutant patients may benefit from CDK4/6 inhibition.
Author Response
Response to reviewer #2 comment
Pan et al.
Reviewer #2:
The Article addressed whether sex and co-mutations impact prognosis of patients with SMARCA4-11 mutated (mutSMARCA4) malignancies, a topic that was previously is not clear. This cohort included patients with next-generation sequencing (NGS) data available.
Male patients had substantially worse OS than female patients both among the entire mutSMARCA4 cohort (HR = 1.71, [95% CI .92-3.18]) with a median OS of 3.0 versus 43.3 months (P < .001), and among the NSCLC subgroup (HR = 14.2, [95% CI 2.76-73.4]) with a median OS of 2.75 months versus un-estimable (P = .02). The reported cohort of patients with mutSMARCA4, males had substantially worse prognosis than females, while mutTP53, mutKRAS, mutCDKN2A and mutSTK11 were differentially associated with prognosis among all patients and among the NSCLC subgroup. The results could suggest potentially unidentified mechanisms that underly this sex and co-mutation-dependent prognostic disparity among patients whose tumor bears a mutSMARCA4.
The study is well written and overall ready for publication.
Response: We thank the reviewer for the excellent review and constructive comment.
Comments:
- The sample sizes are small (especially tumor when separated by tumor type). Would it be possible to extrapolate from public data ( such as the TCGA) to further look at these co-alterations and sex disparities?
Response: We have added Supplemental Figure 1 and the following paragraph with the TCGA dataset analysis in the Result section and discussion in the bottom of the second paragraph in the Discussion section: “We downloaded and performed analysis of all TCGA cases (total 10,182 cases). There were 448 (4.5%) cases that had SMARCA4 mutation/variant. Among them 34 cases had advanced stage, 13 were female patients and 21 were male patients and there was no significant OS difference between male and female patients. However, the median OS for both male and female patients was more than 30 months (Supplemental Figure 1), which is very different than our cohort and the large cohort that was previously reported.” In the Discussion section: “The small TCGA cohort of patients with SMARCA4 mutation/variant likely does not represent the typical patients with SMARCA4-mutated malignancies. It is possible that some of the SMARCA4 variants that were detected were not pathogenic. It is also possible different SMARCA4 mutation/variant and different histology confer different prognosis as reported by Fernando et al. (Fernando TM, Piskol R, Bainer R, et al. Functional characterization of SMARCA4 variants identified by targeted exome-sequencing of 131,668 cancer patients. Nat Commun. 2020;11: 5551.)
- SMARCA4-mutated male patients had substantially dramatic worse prognosis than female patients among all patients with mutSMARCA4 and among the NSCLC subgroup. The mechanistic aspect of sex disparity should be further discussed, whether it may be lifestyle, hormonal or X-Chromosome linked.
Response: This is an excellent suggestion. We have added additional discussion under the Discussion section in the second paragraph.
- Mutant p53 was higher among males (again can this be seen in larger public data sets)
Response: Among all patients with our dataset, male patients indeed had higher percent of TP53 mutation. This is likely related to higher percent of male patients who were smokers. Please see Table 2. The TCGA dataset does not provide smoking history. We have performed analysis of 448 cases with SMARCA4 mutation/variant, only 34 cases had advanced stage malignancy. Please see Supplemental Figure 1.
4) Manuscript “ SMARCA4 loss is synthetic lethal with CDK4/6 inhibition in non-small cell lung cancer” (PMID: 30718506) is relevant and should be discussed in discussion. This study suggests that SMARCA4 mutant patients may benefit from CDK4/6 inhibition.
Response: This is an excellent suggestion. We have incorporated this reference (ref. 48) and added additional discussion (It is possible that the biological actions of mutTP53 and mutKRAS led to altered tumor microenvironment that is more sensitive to treatment in tumors that carried a SMARCA4 mutation. The study by Xue et al. showed that expression of cyclin D1 (CCND1) was reduced in patients with SMARCA4-mutated lung cancer which led to synthetic lethality with CDK4/6 inhibition. Despite overall inhibition of CDK4/6 has not been shown to be effective in lung cancer, it is possible the subset of patients with a SMARCA4 mutation may be more sensitive to CDK4/6 inhibitors.) (please see the bottom of the 3rd paragraph under Discussion section). Thank you.
Reviewer 3 Report
The manuscript entitled “Sex- and Co-mutation-Dependent Prognosis in Patients with 2 SMARCA4-Mutated Malignancies” by Sex- and Co-mutation-Dependent Prognosis in Patients with 2 SMARCA4-Mutated Malignancies was well written and presented, however, few concerns about the manuscript are given below. Given these suggestions the manuscript requires minor revisions.
- What all are the including and excluding criteria for recruiting patients for this study?
- What might be the reason behind the low survival for male patients
- Apart from TP53 mutation, what all are other mutations might play a significant role in this issue, If possible include its mechanism
- The Genomic Landscape of SMARCA4 Alterations and Associations with Outcomes in Patients with Lung Cancer - PubMed (nih.gov) https://pubmed.ncbi.nlm.nih.gov/32709715/- by Adam J Schoenfeld reported that “ KEAP1 had the strongest association with SMARCA4-mutant tumors compared with SMARCA4 wild-type tumors”, however, keap1 was not included, it might be the strong support to results
- Page 8, line 232 “Our study suggests 232 that mutp53 may be associated with a different mechanism in influencing prognosis 233 among patients with a mutSMARCA4.”- It could be better if the authors explain in detail.
- Figures 1b, 1c & 2b could have been given with good resolution and clearer picture.
Author Response
Response to reviewer #3 comment
Pan et al.
Reviewer #3:
The manuscript entitled “Sex- and Co-mutation-Dependent Prognosis in Patients with 2 SMARCA4-Mutated Malignancies” was well written and presented, however, few concerns about the manuscript are given below. Given these suggestions the manuscript requires minor revisions.
Response: We thank the reviewer for this favorable comment.
- What all are the including and excluding criteria for recruiting patients for this study?
Response: We included all patients who had a SMARCA4-mutated malignancy that was detected by the routine StrataNGS testing for advanced malignancies. We have added one sentence under Demographics in Result section to indicate this.
- What might be the reason behind the low survival for male patients
Response: The mechanism is not entirely clear but could be partially related to worse PS and CCI due to higher percent of male patients with history of smoking in the entire cohort. We have added additional discussion on this (In our entire cohort of patients, male patients appear to have worse PS and CCI and higher percent with history of smoking. The smoking lifestyle likely caused male patients to have worse PS and CCI at the time of diagnosis of SMARCA4-mutated malignancy and was partially responsible for their worse OS. Though this correlation was not as evident in the patients with NSCLC only. Other unidentified mechanisms likely exist.). Please see second paragraph under Discussion section.
- Apart from TP53 mutation, what all are other mutations might play a significant role in this issue, If possible include its mechanism.
Response: Other mechanisms are likely and remain to be identified. Our data showed that both CDKN2A, STK11, and Keap1 mutations were associated with worse survival, however, the prevalence of these two mutations were not different between male and female patients.
- The Genomic Landscape of SMARCA4 Alterations and Associations with Outcomes in Patients with Lung Cancer - PubMed (nih.gov) https://pubmed.ncbi.nlm.nih.gov/32709715/- by Adam J Schoenfeld reported that “ KEAP1 had the strongest association with SMARCA4-mutant tumors compared with SMARCA4 wild-type tumors”, however, keap1 was not included, it might be the strong support to results.
Response: We have added Keap1 mutation into the Cox model. Keap1 mutation was associated with worse prognosis, consistent with the Schoenfeld et al. study. Please see Figure 2A in the Result section and the bottom of 3rd paragraph in the Discussion section. We did reference Schoenfeld et al article in our manuscript (Ref. 21).
- Page 8, line 232 “Our study suggests that mutp53 may be associated with a different mechanism in influencing prognosis among patients with a mutSMARCA4.”- It could be better if the authors explain in detail.
Response: We have added further discussion with two additional references. Please see 3rd paragraph under Discussion section: “Our study suggests that mutp53 may be associated with a different mechanism in influencing prognosis among patients with a mutSMARCA4. Such a mechanism may operate differently depending on different histology type. It may cause accelerated progression of most malignancies but not NSCLC. Previous studies showed that SMARCA4 knockdown caused replicative stress and decreased p21 expression by reducing the binding of p53 to its promoter, suggesting that functional loss of both SMARCA4 and p53 may increase tumor vulnerability to therapy.38, 39”.
- Figures 1b, 1c & 2b could have been given with good resolution and clearer picture.
Response: We have replaced these figures with that of higher resolution.